# Contrastive Deterministic Autoencoders For Language Modeling

**Amur Ghose** and **Pascal Poupart**
David R. Cheriton School of Computer Science
University of Waterloo, Waterloo, Canada
Vector Institute, Toronto, Canada
a3ghose@uwaterloo.ca, ppoupart@uwaterloo.ca

## Abstract

Variational autoencoders (VAEs) are a popular family of generative models with wide applicability. Training VAEs, especially for text, often runs into the issue of posterior collapse, resulting in loss of representation quality. Deterministic autoencoders avoid this issue, and have been explored particularly well for images. It is however unclear how to best modify a deterministic model designed for images into a successful one for text. We show that with suitable adaptations, we can significantly improve on batch-normed VAEs (BN-VAEs), a strong benchmark for language modeling with VAEs, by replacing them with analogous deterministic models. We employ techniques from contrastive learning to control the entropy of the aggregate posterior of these models to make it Gaussian. The resulting models skip reparametrization steps in VAE modeling and avoid posterior collapse, while outperforming a broad range of VAE models on text generation and downstream tasks from representations. These improvements are shown to be consistent across both LSTM and Transformer-based VAE architectures. Appropriate comparisons to BERT/GPT-2 based results are also included. We also qualitatively examine the latent space through interpolation to supplement the quantitative aspects of the model.

## 1 Introduction

The variational autoencoder (Kingma and Welling, 2013) - henceforth referred to simply as the VAE - is a classical neural model that utilizes a paired encoder-decoder structure. For every data instance $\mathbf{x_i}$, the encoder network in the VAE is responsible for creating a compressed code distribution $P(\mathbf{z_i}|\mathbf{x_i})$ parametrically. The decoder network, in turn, uses this $P(\mathbf{z_i}|\mathbf{x_i})$ to form an approximation of the real input, $\hat{\mathbf{x}}_i$ through an intermediate sampling step. By minimizing a reconstruction loss between $\mathbf{x_i}$ and $\hat{\mathbf{x}}_i$, along with a KL-divergence

between $P(\mathbf{z_i}|\mathbf{x_i})$ and the isotropic Gaussian distribution, VAEs can perform both generative and denoising (reconstructive) tasks. Minimization of the KL loss allows the VAE to create an isotropic Gaussian distribution to sample and decode from. However, note that the isotropic Gaussian is not required and in NLP researchers have considered training the latent distribution (Fang et al., 2019) and learning structured, discrete representations (Zhao et al., 2018b).

One of the most pressing problems in practical VAE training arises when the encoder's distribution collapses to the standard Gaussian for every instance, that is, $P(\mathbf{z_i}|\mathbf{x_i})$ becomes $\approx \mathcal{N}(\mathbf{0}, \mathbf{I}) \ \forall i$. This problem is commonly termed as **posterior collapse** (Bowman et al., 2016; Razavi et al., 2019; Lucas et al., 2019; He et al., 2019) and lies at the heart of modern issues with VAEs. Many fixes have been proposed towards this problem, ranging from setting a lower bound on the KL term ($\delta$-VAE) (Razavi et al., 2019), aggressively training the encoder and analyzing the 'lag' between the encoder and the decoder (Agg-VAE) (He et al., 2019), force meaningful usage of the code $\mathbf{z_i}$ through skip connections (Skip-VAE) (Dieng et al., 2019). These issues worsen when the VAE employs an autoregressive structure, such as for text or videos (Fang et al., 2019; Zhao et al., 2018b; Dai et al., 2020; Long et al., 2019). Thus, mitigating posterior collapse in VAE architectures **is likely to have outsized benefits in NLP**.

Independent of the VAE model, there exists the idea of aligning the aggregate posterior. The aggregate posterior consists of the latent space distribution, i.e. the distribution over $\mathbf{z}$ formed by evaluating $\int P(\mathbf{x_i})P(\mathbf{z_i}|\mathbf{x_i})$ over all $\mathbf{x_i}$, where the distribution over $\mathbf{x_i}$ is usually computed by an empirical average over the training set. In these methods, the individual distributions, $P(\mathbf{z_i}|\mathbf{x_i})$ may even be Dirac distributions, i.e. the mapping between $\mathbf{z_i}$ and $\mathbf{x_i}$ is purely deterministic. We can still find,

in aggregate, a distribution over $\mathbf{z}$ that is close to an isotropic Gaussian. Methods in this vein utilize Wasserstein-based optimal transport (Tolstikhin et al., 2017), maximum mean discrepancy (Kolouri et al., 2018), etc. for this purpose. These methods cannot truly be termed VAEs, as they are often deterministic, but they work similarly. Due to their deterministic nature and differing loss functions, posterior collapse does not usually occur. In VAEs, the quantity to be maximized is the log likelihood $\log P(\mathbf{x_i})$. This proves intractable, and an equivalent optimization is done via the ELBO (Evidence Lower BOund) objective. While VAEs can be evaluated by log likelihood, another metric is to evaluate their reconstruction error, as well as the quality of samples generated when, in place of $P(\mathbf{z})$, an isotropic Gaussian $\mathcal{N}(\mathbf{0}, \mathbf{I})$ is substituted and the Gaussian samples fed through the decoder network. This notion of sample-based evaluation can also be done for deterministic autoencoders, which do not allow an ELBO evaluation. This allows us to compare deterministic autoencoders to variational options fairly to determine superiority.

## 1.1 Our Contributions

We seek to draw parallels to findings in image datasets that indicate deterministic autoencoding models outperform variational ones in terms of sample quality (Ghosh et al., 2019). Due to their relative freedom from posterior collapse, and the aforementioned outsized impact of posterior collapse in language modeling, adapting these deterministic architectures to NLP may improve autoencoders. We find that replacing a previous highly performing architecture - the BN-VAE (Zhu et al., 2020) with a similar deterministic variant improves the performance in language modeling tasks. We engage in an information theoretic analysis of the constant variance VAE, demonstrating that this case allows mutual information maximization. To aid convergence, we add an entropic regularization through contrastive learning (He et al., 2020). To our knowledge, this linkage of contrastive learning to entropic maximization is novel, though entropic regularization has previously been utilized for deterministic autoencoders (Ghose et al., 2020). We evaluate using perplexity-based benchmarks in both forward and reverse directions (Zhao et al., 2018a) and test the accuracy of our formulation using relatively large autoencoders built using transformer encoder-decoder pairs. In all cases, we

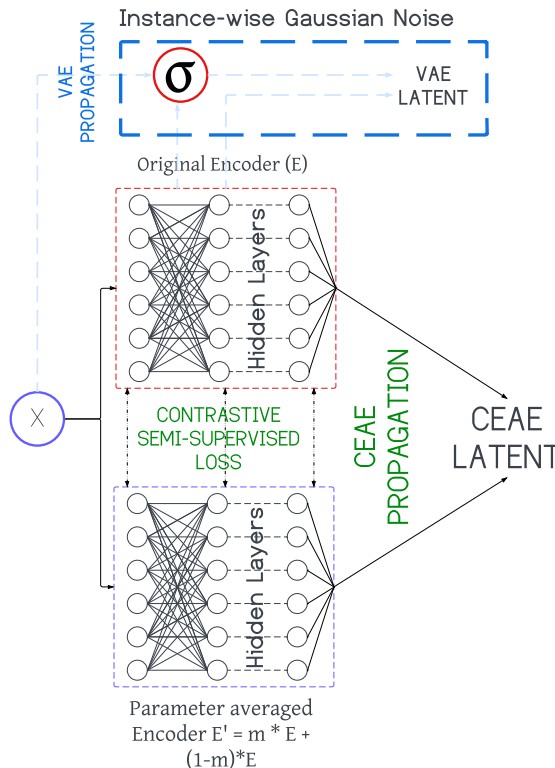

Figure 1: CEAE (Contrastive Entropic Autoencoder) architecture replacing instance-specific noise with contrastive loss. Constant noise (not shown) is annealed over training to make the latent $Z_{CEAE}$ deterministic.

observe improvements not only over the previous BN-VAE, but over a broad array of VAE architectures. We term our architecture CEAE (Contrastive Entropic Autoencoder) in figure 1. **To account for the increasing relevance of large language models, we also test against appropriate architectures that utilize BERT and GPT-2 as parts of an overall VAE architecture.**

## 2 Definitions and Prior Work

### 2.1 Variational Autoencoder

We present here first the classical set up of the VAE. We denote $\mathcal{D}, \mathcal{E}$ as the decoder, encoder halves of the VAE. Generally, $\mathcal{D}, \mathcal{E}$ are comparable in size, shape, architecture, etc. We propagate every data instance $\mathbf{x_i}$ through subnetworks $\mathcal{E}_\mu, \mathcal{E}_{\sigma^2}$:

$$\boldsymbol{\mu_i} = \mathcal{E}_{\boldsymbol{\mu}}(\mathbf{x_i}), \boldsymbol{\sigma_i^2} = \mathcal{E}_{\boldsymbol{\sigma^2}}(\mathbf{x_i}) \qquad (1)$$

These parameters, in turn, define a Gaussian distribution, from which $\mathbf{z_i}$ is drawn.

$$\mathbf{z_i} \sim \mathcal{N}(\boldsymbol{\mu_i}, \boldsymbol{\sigma_i^2}) \qquad (2)$$

The decoder is then used to produce an (approximate) reconstruction $\hat{\mathbf{x}}_\mathbf{i}$ of $\mathbf{x}_\mathbf{i}$:

$$\hat{\mathbf{x}}_\mathbf{i} = \mathcal{D}(\mathbf{z}_\mathbf{i}) \qquad (3)$$

The loss function for VAE optimization is:

$$\frac{1}{2}||\hat{\mathbf{x}}_\mathbf{i} - \mathbf{x}_\mathbf{i}||^2 + KL(\mathcal{N}(\boldsymbol{\mu}_i, {}^2_\mathbf{i})||\mathcal{N}(\mathbf{0}, \mathbf{I})) \qquad (4)$$

where the $||$ sign between the two normal distributions denotes the KL-divergence. Note also that the first term (squared loss) is commonly used in papers that describe VAEs, but it should be replaced by cross entropy in classification tasks (including most NLP tasks). Furthermore, the loss function may not be the same for all variants of the VAE. For instance, the $\beta$-VAE (Higgins et al., 2016) performs the optimization based on

$$\frac{1}{2}||\hat{\mathbf{x}}_\mathbf{i} - \mathbf{x}_\mathbf{i}||^2 + \beta KL(\mathcal{N}(\boldsymbol{\mu}_i, \boldsymbol{\sigma}_i^2)||\mathcal{N}(\mathbf{0}, \mathbf{I})) \quad (5)$$

$\beta$ turns into a tunable hyperparameter, and is generally not 1. Such tuning may improve performance - in particular avoiding posterior collapse.

## 2.2 Deterministic Autoencoders

Utilizing the same notation as the VAE, if we instead consider replacing the sampling step

$$\mathbf{z}_\mathbf{i} \sim \mathcal{N}(\boldsymbol{\mu}_i, \boldsymbol{\sigma}_i^2) \qquad (6)$$

with $\mathbf{z}_\mathbf{i} = \boldsymbol{\mu}_i$, the resulting flow is completely deterministic. In this case, training minimizes:

$$\frac{1}{2}||\hat{\mathbf{x}}_\mathbf{i} - \mathbf{x}_\mathbf{i}||^2 + R(\mathbf{z}_\mathbf{i}) \qquad (7)$$

where $R$ can be any regularizer on $\mathbf{z}$, with the simplest form being a least-squares regularizer, i.e. $R(\mathbf{z}) = ||\mathbf{z}||^2$, forming a **Regularized autoencoder** (RAE) (Ghosh et al., 2019). To generate samples from the RAE, a Gaussian, or a Gaussian mixture model, is fit to the distribution of $\mathbf{z}$ after training. With a suitable regularizer $R$, such as the MMD regularizer forming the Wasserstein autoencoder (WAE) (Tolstikhin et al., 2017), guarantees can be made on the latent space distribution $P(\mathbf{z})$ that do not require this post hoc fitting and allow drawing samples from $\mathcal{N}(\mathbf{0}, \mathbf{I})$ directly - this case is our focus of interest. For image datasets such as MNIST, CIFAR-10, and CelebA, deterministic autoencoders have a notable advantage (Ghosh et al., 2019) and we would like to carry this over to text.

## 2.3 VAE Optimization for Language Modeling

The failure mode in VAE optimization (when modeling text, images or general datasets) manifests itself by posterior collapse, where $\mathcal{N}(\mathbf{0}, \mathbf{I})$ takes the place of every latent code's distribution. Autoregressive VAEs suffer the worst in this matter, which **impacts NLP disproportionately** as non-autoregressive models are generally not usable. It has been suggested that a primary cause of the collapse is due to training dynamics (He et al., 2019). In this view, the inference network (encoder) is in terms of training initially at a very poor performance, which causes the generator network (decoder) to neglect the latent codes generated by the encoder. This in turn leads to uninformative codes (the posterior collapse phenomenon). Alternative fixes include looking at weighing of the KL-term. Such methods include the $\beta$-VAE which adds a weight of $\beta$ to the KL term (Higgins et al., 2016), and methods which do not allow the KL to go to zero (Razavi et al., 2019). Architecture-wise, skip-connections reduce posterior collapse (Dieng et al., 2019), as does reducing the complexity of the decoder network. During the main training loop, the loss can be amortized (Kim et al., 2018), annealed, or applied in a cyclic manner (Fu et al., 2019), all of which reduce the phenomenon. Finally, the optimizer itself may be changed, with SGD being somewhat preferable to ADAM (Srivastava and Sutton, 2017) for the purposes of avoiding posterior collapse. Orthogonal to all these fixes, we may of course simply use deterministic autoencoders.

## 2.4 BN-VAE Architecture - a Strong Baseline

Let $\mu_{ij}, \sigma_{ij}$ denote the $j$-th index of the posterior parameters of the $i$-th instance of a latent code of dimension $K$. It may be shown that the expectation of the KL divergence obeys the relation (Zhu et al., 2020):

$$E[KL] = \frac{1}{2}\sum_{j=1}^{K} E[\mu_{ij}^2] + E[\sigma_{ij}^2] - E[\log \sigma_{ij}^2] - 1 \tag{8}$$

where the expectation is taken over the samples i.e., over $i$. We directly use the resulting inequality:

$$E[KL] \geq \frac{1}{2}\sum_{j=1}^{K} E[\mu_{ij}^2]$$

Because $E[\sigma_{ij}^2] - E[\log \sigma_{ij}^2] - 1 \geq 0$ as $e^x - x \geq 1$. We know that for any random variable $X$ that

has a defined first and second moment, $E[X^2] = Var(X) + (E[X])^2$. We enforce a batch norm constraint on $\mu_{ij}$, that fixes the expectation and/or the variance, thereby setting the lower bound on the expectation of KL. Batch normalization here simply refers to the near-ubiquitous practice of minibatch-level normalization, i.e., adding a layer of the form:

$$\mathbf{BN}(\mathbf{x_i}) = \frac{\mathbf{x_i} - \boldsymbol{\mu}}{\boldsymbol{\sigma}} \qquad (9)$$

where $\boldsymbol{\mu}, \boldsymbol{\sigma}$ represent the mean and standard deviation computed over the minibatch of $\mathbf{x_i}$. This batch norm layer is usually accompanied by an affine function, i.e., a function of the form $f(\mathbf{x}) = \mathbf{Ax} + \mathbf{b}$ (Ioffe and Szegedy, 2015). We will explicitly make a distinction between the two parts. Over experiments against other architectures such as Skip-VAE and others (all of which are purported designs to circumvent the posterior collapse issue) BN-VAE results in superior performance in both language modeling (NLL metrics) and the downstream usage of learnt representations (Zhu et al., 2020). **Our lesson from this success is to set a baseline with a VAE which batch normalizes the means of latent representations.**

## 3 Information Theory of Constant Variance Batch Normed Autoencoders

Let the generation process of the latent code be :

$$z_{ij} \sim \mathcal{N}(\mu_{ij}, \sigma_{ij}^2)$$

With the batch norm constraint that :

$$E[\mu_{ij}] = a, Var[\mu_{ij}] = b \geq 0$$

Consider an intermediate case between the VAE (where each $\sigma_j$ can be distinct) and the deterministic autoencoder (each $\sigma_j = 0$). Set every $\sigma_j = c$, $c > 0$ constant. Then, indexwise and vectorwise,

$$z_{ij} = \mu_{ij} + c, \quad \mathbf{z_i} = \boldsymbol{\mu_i} + Z$$

where $Z$ is a random vector from $\mathcal{N}(\mathbf{0}, c^2\mathbf{I})$. Consider the mutual information between $\mathbf{z}, \boldsymbol{\mu}$. This denotes the amount of useful information sent through the encoding and maximizing it avoids posterior collapse (Zhao et al., 2017). It equals

$$H(\mathbf{z}) - H(\mathbf{z}|\boldsymbol{\mu})$$

Where $H$ is the entropy function. Now, $\mathbf{z}|\boldsymbol{\mu} = (\boldsymbol{\mu} + Z)|\boldsymbol{\mu}$. Therefore, the required entropy is of

$$H(\boldsymbol{\mu} + Z|\boldsymbol{\mu}) = H(Z|\boldsymbol{\mu})$$

$Z$ is independent of $\boldsymbol{\mu}$. This differs from general VAEs, where the variance is instance dependent and $\boldsymbol{\sigma_i}, \boldsymbol{\mu_i}$ are related, relating $Z, \boldsymbol{\mu}$. Note that here we refer to the instance-index $i$ and not the dimension index $j$ as we discuss instance-level dependence. This yields the final mutual information expression as :

$$H(\mathbf{z}) - H(Z)$$

Now, $H(\mathbf{z})$ has a fixed value of $E[\mathbf{z}]$ and also a fixed variance, since it is the direct sum of two random variables $Z$ and $\boldsymbol{\mu}$, both of which, by hypothesis, have fixed means and variances. Under this condition of fixed mean and variance, it is known that the entropy $H$ is maximized iff $\mathbf{z}$ is distributed as a Gaussian (Thomas and Cover, 1999). Therefore, a **more informative constant variance batch normed VAE** induces a more Gaussian representation on $\mathbf{z}$. Since $\boldsymbol{\mu} = \mathbf{z} - Z$, and $\mathbf{z}$ approaches a Gaussian, while $Z$ is itself one, the desired latent mean $\boldsymbol{\mu}$ also approaches a Gaussian as mutual information rises. Note that our analysis holds for any $c > 0$. This means even very low values of $c$ - slowly annealed to $0$, approximating the deterministic autoencoder case - will work as long as the mutual information required is high. To create an aggregate posterior which is the isotropic Gaussian, we assume $a = 0, b = 1$, i.e.

$$E[\boldsymbol{\mu_i}] = \mathbf{0}, Var[\boldsymbol{\mu_i}] = \mathbf{1}$$

When $\boldsymbol{\mu_i}$ becomes a Gaussian with the above two constraints, our job of creating an appropriate deterministic autoencoder with the right aggregate posterior is done. We have already discussed the interaction of mutual information with that process, but now, observe that becoming a Gaussian can also be done by controlling the entropy $H$, which is maximized iff $\boldsymbol{\mu_i}$ is Gaussian, which also implies $\mathbf{z_i}$ is Gaussian. The process accelerates when minimizing a regularizer in the form of $-\lambda H(\boldsymbol{\mu_i}), \lambda \geq 0$.

### 3.1 Entropy and Contrastive Regularization

We require an effective entropic estimator for $H(\boldsymbol{\mu_i})$ to make it Gaussian. A first step may be repulsion-based estimators such as the Kozachenko-Leonenko estimator : (Delattre and Fournier, 2017) for a sample consisting of

$X_1, X_2, \ldots, X_{N+1} \in \mathbb{R}^d$ drawn from an unknown distribution $P$, assuming $N > 1$, With $R_i = \min_{j \neq i} ||X_i - X_j||_2$, $Y_i = N(R_i)^d$, $B_d$ the volume of the unit ball in $\mathbb{R}^n$, $\gamma$ the Euler-Mascheroni constant $\approx 0.577$, an estimate of the entropy of the distribution is:

$$H(P) \approx \frac{1}{N+1} \sum_{i=1}^{N+1} \log Y_i + \log B_d + \gamma$$

The estimator relies primarily on the leading sum over $Y_i$, which computes "repulsions" between latent representations which are too close. Only this sum (with weight $\lambda$) needs to be computed at all for gradient based learning, as the other terms are constants. In practice, this estimator has been used only sporadically for image autoencoders (Ghose et al., 2020) and rarely in general for neural networks, and direct implementations of the method for language autoencoders leads to convergence failure.

We turn to the contrastive learning literature to look for a solution, which has recently emerged as a strong baseline in representation learning for both unsupervised (Chen et al., 2020; Zhang et al., 2022) and supervised (Khosla et al., 2020; Zhang et al., 2022) contexts. In **unsupervised** contrastive learning, it is desired to learn two alternative representations $Z_i, Z_i^+$ of some instance $X_i \in \mathcal{X}$, so that $Z_i, Z_i^+$ are close (e.g. by the inner product) and $Z_j$ arising from $X_j \in \mathcal{X}, j \neq i$ has a low inner product. This is done by minimizing:

$$-\sum_{X_i \in \mathcal{X}} \log \frac{\exp(\langle Z_i, Z_i^+ \rangle)}{\sum_{X_j \in \mathcal{X}, j \neq i} \exp(\langle Z_i, Z_j \rangle)}$$

$Z_i^+$ arises from a noisy or augmented version of $X_i$, such as a crop (if $X$ is an image). One suitable method is momentum contrast (MoCo) (He et al., 2020), where $Z_i$ is generated by a model with parameters $\theta$, and a model with $\theta'$, a time average of $\theta$, generates $Z_i^+$. Therefore, the encoding method learns to be insensitive to these noises and learns encodings $Z_i, Z_i^+$ that are more or less invariant to such. Simultaneously, the denominator discourages proximity between codes $Z_i, Z_j$ arising from different instances $X_i, X_j$ - a repulsive regularization controlling the entropy which sketches our derivation. **Unlike directly controlling the entropy with a Kozachenko-Leonenko repulsion loss**, this method is well understood empirically in terms of training and implementation. **Further, it can be shown that this loss approximates the entropic regularizer for Gaussian distributions in high dimensions**. The full proof and derivation appears in the appendix. We use the following loss function $\mathcal{L}_{ent}$, with $\mu_i, \mu_i^+$ being respectively generated by $\mathcal{E}_\theta, \mathcal{E}_{\theta'}$, where $\theta' = m\theta' + (1-m)\theta$ is a time averaged version of the main model $\mathcal{E}_\theta$.

$$\mathcal{L}_{ent} = -\lambda_t \times \sum_{x_i \in \mathcal{X}} \log \frac{\exp(\langle \mu_i, \mu_i^+ \rangle)}{\sum_{x_j \in \mathcal{X}, j \neq i} \exp(\langle \mu_i, \mu_j \rangle)}$$

The details of how to choose $m, \lambda_t$ and their justifications also appear in the appendix. The overall training loss is formed by adding $\mathcal{L}_{ent}$ to the reconstruction loss $\mathcal{L}_{rec} = \frac{1}{2}||\hat{\mathbf{x}}_\mathbf{i} - \mathbf{x}_\mathbf{i}||^2$.

## 4 Experimental Details and Methodology

### 4.1 Dataset Choices

We present results primarily using the Yahoo and Yelp corpora (Yang et al., 2017) for language modeling tasks using LSTM encoder-decoder architecture autoencoders. This maintains consistency with the BN-VAE (Zhu et al., 2020) in terms of comparing performance for these tasks. We additionally use a small-scale transformer, with its structure based on the Transformer-XL model (Dai et al., 2019) specifically for Penn Tree Bank(PTB) dataset (Marcus et al., 1993), for results on this dataset (in appendix). Unlike transformer-XL, which is a decoder-only model, we employ an encoder-decoder transformer, but keep hyperparameter and training recommendations aligned with the original source code for the PTB Transformer-XL details. We name our architecture **CEAE** - Contrastive Entropic Autoencoder.

### 4.2 Metrics

Generally, variational autoencoders are compared on the following metrics:

- Negative log-likelihood of the data (usually estimated via ELBO/importance sampling)

- Mutual information between $\mathbf{x_i}, \mathbf{z_i}$, capturing latent code quality(Alemi et al., 2016)

- KL divergence (averaged) between each latent code's distribution and the isotropic Gaussian, i.e. $(\mathcal{N}(\boldsymbol{\mu_i}, \boldsymbol{\sigma_i^2}) || \mathcal{N}(\mathbf{0}, \mathbf{I}))$

Other metrics such as active units (AU) (Burda et al., 2015) may be used, which capture the number of latent dimensions which are actually being utilized in the autoencoder. None of the above mentioned metrics (except the AU) can be used to compare VAEs with deterministic autoencoders. We hence use a different suite of metrics based on forward and reverse perplexities (Zhao et al., 2018a; Gagnon-Marchand et al., 2019):

- Reconstruction error, measured by negative log likelihood of decoder-reconstructed input.

- Forward perplexity, where we generate and decode 1000 samples from $\mathcal{N}(\mathbf{0}, \mathbf{I})$. The perplexity of this sample is evaluated by a critic model, optionally after training on the corresponding train segment of the corpus.

- Reverse perplexity, in which $10,000$ samples are generated from the autoencoder just as above, but are now used for training a different model, then tested on the **test** segment of the corresponding corpus.

We chose two different critic models to reflect two different ends of the model power spectrum: a simple LSTM model for language modeling, and GPT-2 (Radford et al., 2019) (standard configuration, 117M parameters). The reverse-perplexity task was performed only with the LSTM critic, as it was found that training GPT-2 on the (relatively low quality compared to real text) samples hurt downstream performance on the uncorrupted test corpus. We add human evaluation of generated samples as a sanity check.

### 4.3 Comparisons and Benchmarking

We include a full suite of comparisons for the Yahoo and Yelp corpora, with the following architectures targeting the posterior collapse problem:

- Skip-VAE: latent-decoder skip connection (Dieng et al., 2019)

- Annealing the KL loss (Bowman et al., 2015)

- $\beta$-VAE (KL weight $\beta$) (Higgins et al., 2016)

- Cyclic annealing of KL loss (Fu et al., 2019)

- Free bits in the KL loss (Kingma et al., 2016)

- $\delta$-VAE (minimum KL $\delta$) (Razavi et al., 2019)

- Von-mises fischer (vMF) VAE (Xu and Durrett, 2018)

- Semi-amortized (SA) VAE (Kim et al., 2018)

- Aggressive-VAE (He et al., 2019)

These benchmarks correspond to the ones in (Zhu et al., 2020), from which we also find the required reconstruction metrics and BN-VAE variants. We add a vanilla VAE and LSTM language model for baselines. For the PTB dataset, we compare the deterministic autoencoder to a VAE setup optimized analogously to the much larger BERT-GPT-2 VAE in (Li et al., 2020). Only the standard VAE is considered, with KL annealing kept for consistency. **Standard deviations, implementation and architectural details, hyperparameters etc. are in the appendix.**

### 4.4 Large Language Model Comparison

We also add a comparison to OPTIMUS architecture of VAEs (Li et al., 2020) with BERT as the encoder, GPT-2 as the decoder, and pre-training on the Wikipedia dataset (details in appendix). Further, the embeddings learnt by our method were **compared to BERT zero-shot embeddings per sentence** (this result appears in the appendix).

## 5 Results

In terms of text modeling, results appear in Tables 1 and 2. In general, our method outperforms the competitors and if not is close to the top performer. It should be noted that realistically, the LSTM critic's performance relative to GPT-2 is due to the fact the LSTM is trained on the relevant corpus while GPT-2 is tested zero-shot. Even though GPT-2 is a stronger model, the LSTM has more domain knowledge, causing their perplexities to be close. This implies that GPT-2 evaluates the samples based on general knowledge of the English language and plausibility as English sentences (as it is tested zero-shot) while the LSTM evaluates it with emphasis on the domain knowledge (which is the sole train data). Having both perplexities thus evaluates differently, and we perform well on both.

To evaluate the quality of the latent space for downstream tasks, we extract the latent representations in a shortened Yelp dataset following (Shen et al., 2017), along with the labels for a small fraction of the dataset. These labels reflect the nature of the shortened review (positive or negative). We

| | Yahoo | | | | Yelp | | | |
|---|---|---|---|---|---|---|---|---|
| **Model** | **Rec** | **GPT2-F** | **L-F** | **L-R** | **Rec** | **GPT2-F** | **L-F** | **L-R** |
| LSTM-LM | 328.0 | 136.7 | 171.5 | 202.3 | 351.1 | 125.2 | 95.1 | 132.0 |
| VAE | 328.6 | 118.2 | 141.7 | 175.8 | 357.9 | 100.1 | 87.8 | 115.5 |
| $\beta$-VAE (0.4) | 322.4 | 126.4 | 136.3 | 173.9 | 354.0 | 97.4 | 91.8 | 113.6 |
| cyclic | 328.5 | 125.4 | 131.8 | 182.6 | 357.5 | 100.6 | 87.8 | 112.8 |
| Skip-VAE | 326.2 | 130.3 | 138.4 | 175.9 | 355.7 | 95.2 | 91.7 | 109.6 |
| SA-VAE | 322.0 | 124.6 | 139.2 | 177.5 | 353.1 | 96.6 | 86.3 | 119.4 |
| Agg-VAE | 321.0 | 118.9 | 145.7 | 178.6 | 352.1 | 97.5 | 85.0 | 121.7 |
| FB (4) | 326.9 | 129.5 | 134.4 | 172.7 | 355.2 | 92.1 | 82.6 | 109.8 |
| FB (5) | 324.9 | 128.2 | 131.5 | 170.2 | 354.9 | 98.8 | 87.2 | **109.0** |
| $\delta$-VAE (0.1) | 327.5 | 122.0 | 130.7 | 171.8 | 356.6 | 95.5 | 84.2 | 109.4 |
| vMF-VAE (13) | 325.4 | 117.6 | 134.8 | 168.2 | 355.5 | 105.2 | 85.3 | 115.0 |
| BN-VAE (0.6) | 320.5 | 110.6 | 128.6 | 165.4 | 350.5 | 92.2 | 78.2 | 112.7 |
| BN-VAE (0.7) | 318.6 | 109.0 | 124.5 | 168.9 | 346.8 | 90.5 | 80.5 | 109.2 |
| CEAE | **316.7** | **107.6** | **119.8** | **163.1** | **342.0** | **88.1** | **78.0\*** | 110.2 |

Table 1: Language modeling. Reconstruction log loss (Rec), Forward perplexity with GPT-2 (GPT2-F), Forward perplexity with LSTM (L-F), Reverse perplexity with LSTM (L-R) in that order on Yahoo, Yelp. Reverse perplexity with GPT-2 was not meaningful as the fine-tuning is without effect due to pre-existing large corpus in the pretraining phase. **Statistical analysis - standard deviations and confidence intervals etc. - appears in appendix.** * indicates statistically insignificant best method.

| | Yahoo | | | | Yelp | | | |
|---|---|---|---|---|---|---|---|---|
| | **Small-scale transformer results** | | | | | | | |
| **Model** | **Rec** | **G2-F** | **L-F** | **L-R** | **Rec** | **G2-F** | **L-F** | **L-R** |
| VAE | 304.6 | 98.0 | 121.4 | 130.5 | 328.2 | 70.1 | 77.8 | 95.2 |
| $\beta$-VAE (0.4) | 305.2 | 86.2 | 116.9 | 120.2 | 330.2 | 72.4 | 71.9 | 102.8 |
| cyclic | 304.6 | 85.6 | 91.8 | 103.5 | 332.7 | 75.6 | 70.4 | 97.5 |
| FB (7) | 303.2 | 79.6 | 94.9 | 105.1 | 329.4 | 74.6 | **68.5** | 98.1 |
| BN-VAE (0.7) | 301.7 | 76.9 | 102.0 | 112.9 | 328.5 | **65.7** | 70.2 | 96.8 |
| CEAE | **300.2** | **72.5** | **89.2** | **101.6** | **326.1** | 69.7 | 70.9 | **92.1** |
| | **OPTIMUS (BERT encoder, GPT-2 decoder) results** | | | | | | | |
| **Model** | **Rec** | **G2-F** | **L-F** | **L-R** | **Rec** | **G2-F** | **L-F** | **L-R** |
| OPTIMUS | 282.8 | 43.8 | 80.5 | 96.1 | 334.3 | 47.0 | 61.8 | 83.7 |
| BN-VAE (0.7) | 285.8 | 45.5 | 85.4 | 99.2 | 330.5 | 45.8 | 60.4 | 82.5 |
| CEAE | **278.7** | **42.4** | **79.4** | **91.8** | **328.7** | **45.0\*** | **57.2** | **80.6** |

Table 2: Performance of transformer autoencoders on Yahoo and Yelp, evaluated using the same metrics.

then apply a simple shallow neural classifier to get the labels (review valence). Note that the task of representation learning always has access to the same number of unlabeled sentences, only the number of labels is varied. Our method proves superior especially as more labels become available with all results in Table 3. For human evaluation, we gathered five individuals (graduate students or machine learning engineers) who were provided 200 choice-based questions and asked to pick the most coherently generated choice among permuted options. Each choice corresponded to a sample from a method on a corpus. We compare to the BN-VAE and vanilla VAE and find a significant advantage in terms of being chosen across both LSTM and transformer architectures, as shown in Tables 4 and 5. **A follow-up with GPT-4 appears in the appendix.** Overall, across a broad variety of tasks, we improve on the BN-VAE architecture.

| #label | 100 | 500 | 1k | 2k | 10k |
|---|---|---|---|---|---|
| AE | 78.6 | 87.1 | 91.0 | 92.4 | 94.7 |
| VAE | 56.5 | 86.7 | 91.4 | 91.1 | 95.1 |
| $\delta$-VAE | 57.6 | 73.4 | 90.8 | 91.0 | 94.5 |
| Agg-VAE | 72.3 | 82.4 | 90.2 | 91.3 | 94.2 |
| cyclic | 65.4 | 76.4 | 88.5 | 91.1 | 93.7 |
| FB (9) | 74.3 | 87.2 | 90.6 | 91.5 | 95.2 |
| AE+FB (6) | 82.4 | 91.0 | 93.1 | 94.2 | 95.3 |
| BN-VAE (0.6) | 85.6\* | 92.4 | 93.0 | 94.3 | 95.3 |
| CEAE | **85.6\*** | **92.7** | **93.5** | **94.6** | **95.5** |

Table 3: Accuracy on Yelp - downstream task performance, with a small MLP trained on labeled samples of a fixed number. * indicates statistical insignificance.

| Candidate | CEAE | BN-VAE | VAE |
|---|---|---|---|
| Yahoo | **46.2** % | 36.2 % | 17.6 % |
| Yelp | **56.1** % | 30.4% | 13.5 % |

Table 4: Frequency of choice among generated samples among **LSTM** models.

| Candidate | CEAE | BN-VAE | VAE |
|-----------|------|--------|-----|
| Yahoo | **49.8** % | 30.5 % | 19.7 % |
| Yelp | **43.6** % | 33.6 % | 22.8 % |

Table 5: Frequency of choice among generated samples among **transformer** models.

## 6 Qualitative and Quantitative Analysis of Interpolations

Autoencoders allow smooth linear interpolation between two sentences in the latent space in a manner that should allow both **syntactic** (sentence structure) and **semantic** meaning. This is captured using the subset of the short Yelp review dataset, which consists of small single sentence reviews of establishments with either 0 or 1 for a negative or positive review, and is used in the classification task in Table 3. We perform the following sanity checks:

- That interpolating between a positive and a negative review yields a neutral review.

- That interpolating between two reviews of the same nature (positive-positive or negative-negative) always yields reviews of the same nature, but of differing content or sentence structure, reflecting the source sentences.

- That these interpolations have numerical scores (from the classifier of Table 3) that match the decoded content.

Results demonstrating these qualitative characteristics are summarized in Table 6. Moving between reviews of different kinds changes the score as expected to one which is ambivalent i.e. around 0.5, which reflects in the text as well such as in the example on row 3 of the positive and negative interpolation. Between reviews of the same nature (clustered around 0 or 1) interpolation causes changes in sentence structure and content - in the case of two negative reviews, the interpolation closer to the sentence "are you kidding me?" begins with "are you", and the interpolation involving two positive reviews also associates common sentence structures.

## 7 Conclusions and Future Work

Attention based transformer models (Vaswani et al., 2017) have achieved great success in all areas of NLP (Devlin et al., 2018; Radford et al., 2018, 2019) . Transformer models retain an autoencoder-like parallel with encoder and decoder modules. Though they are dissimilar to VAEs in the sense that often, decoder-only models are used for text generation, whereas encoder-only models are used for representation learning, using both simultaneously creates a VAE-like architecture. This analogy has been used to train massive VAEs for NLP e.g. OPTIMUS (Li et al., 2020) that employ pretrained encoders. We view our work as an indication that deterministic autoencoding, unlike traditional VAEs, can design better autoencoders for text. Issues of VAE training, namely posterior collapse, worsen with increasing model power (Yang et al., 2017; Semeniuta et al., 2017; Razavi et al., 2019). Powerful VAE design requires tackling this and deterministic models may offer the solution. We also consider our work to be of significance to the field in its successful usage of contrastive learning for NLP, which for text often suffers from less clear augmentations (Rethmeier and Augenstein, 2021) relative to images. We focus on BatchNorm, however, in natural language tasks, increasing emphasis is being laid on layer normalization aka LayerNorm (Ba et al., 2016; Xu et al., 2019) which forms a key part of Transformers. We discuss the Layer norm case further in the appendix.

## 8 Limitations

Our methodology focuses on autoencoders which may include transformer architectures, however, by necessity, autoencoders involve an encoder-decoder pairing which may be absent in some architectures which may be encoder-only (for word embeddings) or decoder-only (for language generation). In these cases, our approach is not scalable and will require some rethinking. Further, text generation is a field in a state of flux. Although we have tested with large models in the form of GPT-2, it is possible our results do not scale to as-of-yet unreleased but extremely potent models, such as GPT-4.

## 9 Ethics Statement

Text generation may include harmful content, undesirable generation and algorithmic bias. We do not, however, view our work as being particularly prone to these failure modes any more than other papers in this domain, and we believe no particularly strong ethical statement is warranted.

## References

Alexander A Alemi, Ian Fischer, Joshua V Dillon, and Kevin Murphy. 2016. Deep variational information bottleneck. *arXiv preprint arXiv:1612.00410*.

**Interpolation between two negative reviews**

| Type | Text | Score |
|---|---|---|
| Source | i was really disappointed . | 0.22 |
| Interpolation | i was really disappointed with the beans. | 0.20 |
| Interpolation | are you kidding with the salt ? | 0.15 |
| Source | are you kidding me ? | 0.17 |

**Interpolation between two positive reviews**

| Type | Text | Score |
|---|---|---|
| Source | their ranch sauce is the best i ever had . | 0.91 |
| Interpolation | their pizza is the best in madison . | 0.88 |
| Interpolation | i also love the tasty steak . | 0.82 |
| Source | i also love the crunchy tacos and chicken enchiladas . | 0.80 |

**Interpolation between a positive and negative review**

| Type | Text | Score |
|---|---|---|
| Source | the pollo especial is excellent , but my favorite is the fiesta platter . | 0.86 |
| Interpolation | the coffee is solid, and a good first impression. | 0.78 |
| Interpolation | i don't get their burger and there is zero energy . | 0.44 |
| Source | i don't get how they use better ingredients and it tastes terrible . | 0.29 |

Table 6: Interpolation between different reviews using LSTM models, with weights $(0.8, 0.2)$ and $(0.2, 0.8)$ respectively for the first/second interpolations relative to source sentences. There is change of estimated review score calculated by the classifier used to compute the accuracy in Table 3 matching the text. Sentences tagged with "Source" are original sentences retrieved as-is from the corpus.

Jimmy Lei Ba, Jamie Ryan Kiros, and Geoffrey E Hinton. 2016. Layer normalization. *arXiv preprint arXiv:1607.06450*.

Arindam Banerjee, Inderjit S Dhillon, Joydeep Ghosh, Suvrit Sra, and Greg Ridgeway. 2005. Clustering on the unit hypersphere using von mises-fisher distributions. *Journal of Machine Learning Research*, 6(9).

Etienne Barnard. 2010. Maximum leave-one-out likelihood for kernel density estimation. Pattern Recognition Association of South Africa and Mechatronics . . . .

Samuel R Bowman, Luke Vilnis, Oriol Vinyals, Andrew M Dai, Rafal Jozefowicz, and Samy Bengio. 2015. Generating sentences from a continuous space. *arXiv preprint arXiv:1511.06349*.

Samuel R Bowman, Luke Vilnis, Oriol Vinyals, Andrew M Dai, Rafal Jozefowicz, and Samy Bengio. 2016. Generating sentences from a continuous space. In *20th SIGNLL Conference on Computational Natural Language Learning, CoNLL 2016*, pages 10–21. Association for Computational Linguistics (ACL).

Yuri Burda, Roger Grosse, and Ruslan Salakhutdinov. 2015. Importance weighted autoencoders. *arXiv preprint arXiv:1509.00519*.

Xinlei Chen, Haoqi Fan, Ross Girshick, and Kaiming He. 2020. Improved baselines with momentum contrastive learning. *arXiv preprint arXiv:2003.04297*.

Shuyang Dai, Zhe Gan, Yu Cheng, Chenyang Tao, Lawrence Carin, and Jingjing Liu. 2020. Apo-vae: Text generation in hyperbolic space. *arXiv preprint arXiv:2005.00054*.

Zihang Dai, Zhilin Yang, Yiming Yang, Jaime Carbonell, Quoc V Le, and Ruslan Salakhutdinov. 2019. Transformer-xl: Attentive language models beyond a fixed-length context. *arXiv preprint arXiv:1901.02860*.

Sylvain Delattre and Nicolas Fournier. 2017. On the kozachenko–leonenko entropy estimator. *Journal of Statistical Planning and Inference*, 185:69–93.

Jacob Devlin, Ming-Wei Chang, Kenton Lee, and Kristina Toutanova. 2018. Bert: Pre-training of deep bidirectional transformers for language understanding. *arXiv preprint arXiv:1810.04805*.

Adji B Dieng, Yoon Kim, Alexander M Rush, and David M Blei. 2019. Avoiding latent variable collapse with generative skip models. In *The 22nd International Conference on Artificial Intelligence and Statistics*, pages 2397–2405. PMLR.

Le Fang, Chunyuan Li, Jianfeng Gao, Wen Dong, and Changyou Chen. 2019. Implicit deep latent variable models for text generation. In *Proceedings of the 2019 Conference on Empirical Methods in Natural Language Processing and the 9th International Joint Conference on Natural Language Processing (EMNLP-IJCNLP)*, pages 3946–3956.

Hao Fu, Chunyuan Li, Xiaodong Liu, Jianfeng Gao, Asli Celikyilmaz, and Lawrence Carin. 2019. Cyclical annealing schedule: A simple approach to mitigating kl vanishing. *arXiv preprint arXiv:1903.10145*.

Jules Gagnon-Marchand, Hamed Sadeghi, Md Akmal Haidar, and Mehdi Rezagholizadeh. 2019. Salsatext: self attentive latent space based adversarial text generation. In *Canadian Conference on Artificial Intelligence*, pages 119–131. Springer.

Amur Ghose, Abdullah Rashwan, and Pascal Poupart. 2020. Batch norm with entropic regularization turns deterministic autoencoders into generative models. In *Conference on Uncertainty in Artificial Intelligence*, pages 1079–1088. PMLR.

Partha Ghosh, Mehdi SM Sajjadi, Antonio Vergari, Michael Black, and Bernhard Schölkopf. 2019. From variational to deterministic autoencoders. *arXiv preprint arXiv:1903.12436*.

Junxian He, Daniel Spokoyny, Graham Neubig, and Taylor Berg-Kirkpatrick. 2019. Lagging inference networks and posterior collapse in variational autoencoders. *arXiv preprint arXiv:1901.05534*.

Kaiming He, Haoqi Fan, Yuxin Wu, Saining Xie, and Ross Girshick. 2020. Momentum contrast for unsupervised visual representation learning. In *Proceedings of the IEEE/CVF Conference on Computer Vision and Pattern Recognition*, pages 9729–9738.

Irina Higgins, Loic Matthey, Arka Pal, Christopher Burgess, Xavier Glorot, Matthew Botvinick, Shakir Mohamed, and Alexander Lerchner. 2016. beta-vae: Learning basic visual concepts with a constrained variational framework.

Sergey Ioffe and Christian Szegedy. 2015. Batch normalization: Accelerating deep network training by reducing internal covariate shift. In *International conference on machine learning*, pages 448–456. PMLR.

S Sathiya Keerthi and Chih-Jen Lin. 2003. Asymptotic behaviors of support vector machines with gaussian kernel. *Neural computation*, 15(7):1667–1689.

Prannay Khosla, Piotr Teterwak, Chen Wang, Aaron Sarna, Yonglong Tian, Phillip Isola, Aaron Maschinot, Ce Liu, and Dilip Krishnan. 2020. Supervised contrastive learning. *Advances in Neural Information Processing Systems*, 33:18661–18673.

Yoon Kim, Sam Wiseman, Andrew Miller, David Sontag, and Alexander Rush. 2018. Semi-amortized variational autoencoders. In *International Conference on Machine Learning*, pages 2678–2687. PMLR.

Diederik P Kingma and Max Welling. 2013. Auto-encoding variational bayes. *arXiv preprint arXiv:1312.6114*.

Durk P Kingma, Tim Salimans, Rafal Jozefowicz, Xi Chen, Ilya Sutskever, and Max Welling. 2016. Improved variational inference with inverse autoregressive flow. *Advances in neural information processing systems*, 29:4743–4751.

Soheil Kolouri, Phillip E Pope, Charles E Martin, and Gustavo K Rohde. 2018. Sliced-wasserstein autoencoder: An embarrassingly simple generative model. *arXiv preprint arXiv:1804.01947*.

Chunyuan Li, Xiang Gao, Yuan Li, Baolin Peng, Xiujun Li, Yizhe Zhang, and Jianfeng Gao. 2020. Optimus: Organizing sentences via pre-trained modeling of a latent space. *arXiv preprint arXiv:2004.04092*.

Teng Long, Yanshuai Cao, and Jackie Chi Kit Cheung. 2019. On posterior collapse and encoder feature dispersion in sequence vaes. *arXiv preprint arXiv:1911.03976*.

James Lucas, George Tucker, Roger B Grosse, and Mohammad Norouzi. 2019. Don't blame the elbo! a linear vae perspective on posterior collapse. *Advances in Neural Information Processing Systems*, 32:9408–9418.

Mitchell Marcus, Beatrice Santorini, and Mary Ann Marcinkiewicz. 1993. Building a large annotated corpus of english: The penn treebank.

Alec Radford, Karthik Narasimhan, Tim Salimans, and Ilya Sutskever. 2018. Improving language understanding by generative pre-training.

Alec Radford, Jeffrey Wu, Rewon Child, David Luan, Dario Amodei, Ilya Sutskever, et al. 2019. Language models are unsupervised multitask learners. *OpenAI blog*, 1(8):9.

Ali Razavi, Aäron van den Oord, Ben Poole, and Oriol Vinyals. 2019. Preventing posterior collapse with delta-vaes. *arXiv preprint arXiv:1901.03416*.

Nils Rethmeier and Isabelle Augenstein. 2021. A primer on contrastive pretraining in language processing: Methods, lessons learned and perspectives. *arXiv preprint arXiv:2102.12982*.

Stanislau Semeniuta, Aliaksei Severyn, and Erhardt Barth. 2017. A hybrid convolutional variational autoencoder for text generation. *arXiv preprint arXiv:1702.02390*.

Tianxiao Shen, Tao Lei, Regina Barzilay, and Tommi Jaakkola. 2017. Style transfer from non-parallel text by cross-alignment. *arXiv preprint arXiv:1705.09655*.

Akash Srivastava and Charles Sutton. 2017. Autoencoding variational inference for topic models. *arXiv preprint arXiv:1703.01488*.

Joy A Thomas and Thomas M Cover. 1999. *Elements of information theory*. John Wiley & Sons.

Ilya Tolstikhin, Olivier Bousquet, Sylvain Gelly, and Bernhard Schoelkopf. 2017. Wasserstein autoencoders. *arXiv preprint arXiv:1711.01558*.

Ashish Vaswani, Noam Shazeer, Niki Parmar, Jakob Uszkoreit, Llion Jones, Aidan N Gomez, Łukasz Kaiser, and Illia Polosukhin. 2017. Attention is all you need. In *Advances in neural information processing systems*, pages 5998–6008.

Roman Vershynin. 2018. *High-dimensional probability: An introduction with applications in data science*, volume 47. Cambridge university press.

Tongzhou Wang and Phillip Isola. 2020. Understanding contrastive representation learning through alignment and uniformity on the hypersphere. In *International Conference on Machine Learning*, pages 9929–9939. PMLR.

Jiacheng Xu and Greg Durrett. 2018. Spherical latent spaces for stable variational autoencoders. *arXiv preprint arXiv:1808.10805*.

Jingjing Xu, Xu Sun, Zhiyuan Zhang, Guangxiang Zhao, and Junyang Lin. 2019. Understanding and improving layer normalization. *arXiv preprint arXiv:1911.07013*.

Zichao Yang, Zhiting Hu, Ruslan Salakhutdinov, and Taylor Berg-Kirkpatrick. 2017. Improved variational autoencoders for text modeling using dilated convolutions. In *International conference on machine learning*, pages 3881–3890. PMLR.

Rui Zhang, Yangfeng Ji, Yue Zhang, and Rebecca J Passonneau. 2022. Contrastive data and learning for natural language processing. In *Proceedings of the 2022 Conference of the North American Chapter of the Association for Computational Linguistics: Human Language Technologies: Tutorial Abstracts*, pages 39–47.

Junbo Zhao, Yoon Kim, Kelly Zhang, Alexander Rush, and Yann LeCun. 2018a. Adversarially regularized autoencoders. In *International conference on machine learning*, pages 5902–5911. PMLR.

Shengjia Zhao, Jiaming Song, and Stefano Ermon. 2017. Infovae: Information maximizing variational autoencoders. *arXiv preprint arXiv:1706.02262*.

Tiancheng Zhao, Kyusong Lee, and Maxine Eskenazi. 2018b. Unsupervised discrete sentence representation learning for interpretable neural dialog generation. In *Proceedings of the 56th Annual Meeting of the Association for Computational Linguistics (Volume 1: Long Papers)*, pages 1098–1107.

Qile Zhu, Jianlin Su, Wei Bi, Xiaojiang Liu, Xiyao Ma, Xiaolin Li, and Dapeng Wu. 2020. A batch normalized inference network keeps the kl vanishing away. *arXiv preprint arXiv:2004.12585*.

## Appendix : contents

In order, we go over :

- The training process and hyperparameters - and how to choose them

- Proof of how MoCo begins to approximate the entropy regularizer

- Additional results on penn tree bank

- GPT-4 validation of human evaluators

- BERT validation of embeddings

- Layer norm discussion

- Confidence intervals and standard deviations of results

## A    Details of Training Process and Hyperparameters

### A.1    Training Flow

From the discussion in section 3.1, it is clear that we will achieve our goal of an aggregate Gaussian-distributed, deterministic $\boldsymbol{\mu_i}$ by either driving up its entropy or requiring high mutual information. Let the $\boldsymbol{\sigma_i} = c$. The overall maximum mutual information equals :

$$H(\mathbf{z}) - H(Z)$$

By applying the Gaussian entropy formula which states that for variance $V$, the entropy of a Gaussian random variable is $\frac{1}{2}\log(2\pi e V)$, we obtain that the mutual information per latent dimension equals

$$\leq \frac{1}{2}\log(2\pi e(1 + c^2)) - \frac{1}{2}\log(2\pi e(c^2))$$

$$= \frac{1}{2}\log(1 + 1/c^2)$$

The upper bound is reached, and the Gaussian distribution formed only when $c$ is high, and the expression is bounded above by a low value which is reached to lower the reconstruction error. We set for epoch $t$ : $c_t = c_0/t, t \leq t_{max}, c_t = 0 \forall t > t_{max}$ - $t_{max}$ is the point where deterministic training begins, i.e. $c = 0$. Depending on the dataset, the value of $c_0$ is **can be chosen with justifications** and our process of choosing it appears in appendix $A.4$. As $c$ goes to zero, the final epochs are trained deterministically, with the entropic loss

$\mathcal{L}_{ent}$ taking over - since the need to raise mutual information arises from the high values of $c_t$, we utilize an epoch dependent regularization on $H(\mu_i)$ minimizing the contrastive loss, unifying notation from $X_i$ to $x_i$

$$\mathcal{L}_{ent} = -\lambda_t \times \sum_{x_i \in \mathcal{X}} \log \frac{\exp(\langle \mu_i, \mu_i^+ \rangle)}{\sum_{x_j \in \mathcal{X}, j \neq i} \exp(\langle \mu_i, \mu_j \rangle)}$$

with $\lambda_t = (1 - 1/t)\lambda_0$, and $\mu_i, \mu_i^+$ being respectively generated by $\mathcal{E}_\theta, \mathcal{E}_{\theta'}$, where $\theta' = m\theta' + (1 - m)\theta$ is a time averaged version of the main model $\mathcal{E}_\theta$. $m \approx 1$ and we use $m = 0.999$ to make $\theta'$ update slowly in keeping with standard implementations (He et al., 2020). The overall training loss is formed by adding $\mathcal{L}_{ent}$ to the reconstruction loss $\mathcal{L}_{rec} = \frac{1}{2}||\hat{\mathbf{x}}_\mathbf{i} - \mathbf{x}_\mathbf{i}||^2$. Due to $\mathcal{L}_{ent}$ in the second and $c$ in the first stage of training, there is no need to fit a post hoc distribution (Ghosh et al., 2019) and the resultant posterior is a Gaussian. Two-stage training is necessary as the contrastive loss only begins to approximate the entropy under assumptions reached later in the training process.

### A.2    Architectural Details

For the LSTM-based VAE for Yahoo and Yelp, architectural details were tuned to match the BN-VAE, with latent dimension of $512$, and single layer LSTMs of size $1024$ for both decoder and encoder, with corresponding feeding patterns of the latent code following exactly the BN-VAE implementation. Minibatches of size 32 with SGD and gradient clipping were used for training, along with annealing on either the KL or the entropy loss following original BN-VAE implementation [1].

For training critic models in the forward and reverse perplexity tasks, a simple recurrent LSTM with two hidden layers of size 200 was used and trained also by SGD. While the LSTM was trained on the text corpus before evaluation on the sample for forward perplexity, GPT-2 was tested zero-shot. Due to the structure of GPT-2, the sample is also evaluated for coherence over all generated sentences, even though each sentence is generated individually. However, we expect differences between our models to respect only the individual quality of generated sentences, as all of them have this independent generation framework. For the Penn Tree Bank models using a transformer model based on Transformer-XL, we built on source code

---

[1] https://github.com/valdersoul/bn-vae

 released from the author's website. The decoder-only model was modified (keeping hyperparameters and latent sizes) to an encoder-decoder model. Architectural details were kept as-is, with 6 layers for both encoding and decoding and using a sentence level representation based on the start of sentence token.

### A.3 Randomness and Seed Dependence, Hardware, Number of Runs

We do not consider our method to depend on randomness i.e. the seed meaningfully. For the BN-VAE architectures we ran with both the pre-provided fixed seed 783435 for the BN-VAE repository and also without setting any seed averaged over 10 runs. The best result among the two was reported (except the reconstruction loss for which we directly use the previous figures reported - we could not actually replicate those numbers in re-running them, however, we got within 0.1 so we consider them correct in the original BN-VAE paper). For our methods, no fixed seed was set and the average over 10 runs was reported directly. Results were reported with a Tesla V100 32 GB, using Pytorch 1.6, on Ubuntu 18.04.

### A.4 Hyperparameter Optimization and Notes on Transformer Training

In general, we keep all hyperparameter and architectural details in line with previous implementations (linked in footnotes). As reported in the paper, the number of cached minibatches is $r = 3$, and every $K = 5$ minibatches we use the global statistics of the batchnorm. Further we try $m = 0.99, 0.999$ for the momentum update step and find better results with 0.99 which leads to the figures reported in the paper, however $m = 0.999$ also outperforms BN-VAE. The SGD is trained with a gradient clip of 5.0 and initial learning rate of 0.5 (BN-VAE) and 0.1 (ours) decayed by a factor of 2 (i.e. multiplied with 0.5) at most 5 times with a decay criterion based on non-improvement on validation for 5 epochs (same as BN-VAE). However, in our experiments, we find that the gradient clip is the only sensitive hyperparameter for both architectures.

The BN-VAE parameters that fix batch norm statistics ($\gamma$) are set according to the original implementation's best performances, at 0.6, 0.7, while for our case the results reported in the paper correspond to setting the contrastive entropic reg-

[2]http://zihangdai.github.io/misc/ptb.zip

ularizer's weight to $6 \times 10^{-3}$. This hyperparameter was based on trying all combinations of $\{2, 3, 6, 7\} \times 10^{\{-3, -4\}}$, i.e. among 8 choices. All entropic hyperparameters with $10^{-3}$ order obtained comparable results to the ones reported in the paper, and with $10^{-4}$ still outperformed BN-VAE. In general, all architectural details, parameters, hyperparameters that do not differ between BN-VAE and our method follow exactly for LSTMs. For transformers, the same KL weight, $r, K, m$ was kept as for LSTMs. However the other parameters were shifted to match the implementation of a small scale transformer for Penn Tree Bank as in Transformer-XL with the exception of changing the decoder-only model to a model using both transformer encoder and decoder layers to more closely match the autoencoder framework.

**Choice of $c_0$ and associated parameters** : We recall from the main text that at epoch zero the initial channel capacity (maximum mutual information) of the autoencoder with $P$ latent units is :

$$\frac{P}{2} \log(1 + 1/c_0^2)$$

For perfect decoding of even the train set consisting of $M$ sentences with lengths $n_i, \ldots, n_M$, we have to consider all valid targets. For a sentence, this is any contiguous subsentence. For a length $n_i$, it equals $\frac{n_i(n_i-1)}{2}$. So we compute the total valid subsegments of the train set as :

$$S = \sum_{i=1}^{M} \frac{n_i(n_i - 1)}{2}$$

By the channel capacity theorem, for perfect reconstruction we would need

$$(1 + 1/c_0^2)^{P/2} \geq S$$

However, this assumes a perfect function approximator on part of the decoder and encoder. As such, we choose $c_0$ to support $(1 + 1/c_0^2)^{P/2} = 5S$.

### A.5 OPTIMUS training

We followed the training procedure from the original paper (Li et al., 2020), taking the lowest $\lambda$ model unless there was a tie in the perplexity values and a better reconstruction was available. Note that we also examined experimental results using different $\lambda$ values. The following figures should be taken in context with respect to Table 2. For Yahoo,

both $\lambda = 0.5, 1$ do obtain better reconstruction accuracy (275.9, 270.8) respectively for OPTIMUS. However, this comes at the cost of significantly worse metrics for G2-F, L-F, and L-R (written as triplets) : (45.2, 84.7, 97.5) for $\lambda = 0.5$ (i.e. statistically comparable to BN-VAE and significantly worse than CEAE) and (47.6, 88.2, 100.2) at $\lambda = 1$ which is worse than the other two models. For Yelp, only $\lambda = 1$ is superior with a reconstruction of 325.8. However, this comes with a G2-F, L-F, L-R triplet of (49.2, 63.1, 84.8) which is again significantly worse than our CEAE result.

For adapting BN-VAE and CEAE, we note that the original training process used a KL thresholding. This was kept as-is. To create smoother training, we train GPT-2 for 1 epoch as a fine-tune step as per OPTIMUS before beginning the main training loop for all of our models.

## B   Relation of MoCo to Entropy Approximation

Our analysis here follows previous theoretical analyses of contrastive learning (Wang and Isola, 2020). We also repeatedly use the property that high dimensional isotropic Gaussians are clustered around a scaled hypersphere, that is, their $L_2$ norm is tightly concentrated. For an exposition on these matters, we refer the reader to (Vershynin, 2018). We use the properties of isotropic Gaussians in high dimensions without further reference. We recall that we use a MoCo loss of the form (with the understanding that $\mu_i$ arises from $X_i$ through an encoder $\mathcal{E}_\theta$ and correspondingly $\mu_i^+$ is from an encoder of parameters $\theta'$ :

$$-\sum_{X_i \in \mathcal{X}} \log \frac{\exp(\langle \mu_i, \mu_i^+ \rangle)}{\sum_{X_j \in \mathcal{X}, j \neq i} \exp(\langle \mu_i, \mu_j \rangle)}$$

Generally, this loss includes the positive pair in the denominator, i.e.

$$-\sum_{X_i \in \mathcal{X}} \log \frac{\exp(\langle \mu_i, \mu_i^+ \rangle)}{\sum_{X_j \in \mathcal{X}} \exp(\langle \mu_i, \mu_j \rangle)}$$

Note that in the general case there is also a temperature hyperparameter $\tau$ which scales the inner products, i.e. $\langle \mu_i, \mu_i^+ \rangle$ becomes $\frac{\langle \mu_i, \mu_i^+ \rangle}{\tau}$ and so on. We ignore this for the sake of exposition and set $\tau = 1$, our methods will carry over to that case as well. Recall that $\mu_i, \mu_i^+$ for us arise from two

distinct encoders respectively both of which receive $X_i$ or $x_i$ (abusing notation), one of which has parameters $\theta$, and the other has parameters $\theta' = m\theta' + (1 - m)\theta$, updated with $m \approx 1$. When near convergence, we may assume that $\mu_i \approx \mu_i^+$, since $\theta' \approx \theta$. Let us now assume for the moment that $\|\mu_i\| = c$, a constant. The above sum for the second case then becomes :

$$-\sum_{X_i \in \mathcal{X}} \log \frac{\exp(c^2)}{\sum_{X_j \in \mathcal{X}} \exp(\langle \mu_i, \mu_j \rangle)}$$

Since $c$ is a constant, the log can be taken out, and $\sum_{X_i \in \mathcal{X}}$ is just an expectation over the dataset. We are left with minimizing

$$\mathbb{E} \log \sum_{X_j \in \mathcal{X}} \exp(\langle \mu_i, \mu_j \rangle)$$

The term within the log, in expectation, is a kernel function (a valid kernel for probability density estimation and thus an estimator of the density). Specifically, it is the (unscaled) vMF (Von Mises Fisher) kernel (Banerjee et al., 2005), which uses the cosine distance on the hypersphere. We then have that

$$\sum_{X_j \in \mathcal{X}} \exp(\langle \mu_i, \mu_j \rangle) = C(d, |\mathcal{X}|) P(\mu_i)$$

Where $C(d, |\mathcal{X}|)$ is a scaling function for the kernel density dependent on the number of elements and the dimension, and $P(\mu_i)$ is the kernel estimate of the probability. The expectation of the log then asymptotically converges to the entropy.

Now let us examine the cases of :

- $\mu_i$ is not of constant norm
- The sum does not include the positive term
- The convergence is not asymptotic

**Varying norm** : In this case, we turn

$$-\sum_{X_i \in \mathcal{X}} \log \frac{\exp(c^2)}{\sum_{X_j \in \mathcal{X}} \exp(\langle \mu_i, \mu_j \rangle)}$$

into

$$-\sum_{X_i \in \mathcal{X}} \log \frac{\exp(\|\mu_i\|^2)}{\sum_{X_j \in \mathcal{X}} \exp(\langle \mu_i, \mu_j \rangle)}$$

Which can be turned into minimizing :

$$-\mathbb{E}\|\mu_i\|^2 + \mathbb{E}\log\sum_{X_j\in\mathcal{X}}\exp(\langle\mu_i,\mu_j\rangle)$$

But by hypothesis, all $\mu_i$ are batch-normed with fixed statistics (0 mean 1 variance). We conclude that the new term introduced is a constant, and cannot influence optimization.

More pressingly, we are no longer on the hypersphere, and cannot use the vMF kernel without checking it works. However, note that the vMF kernel, i.e.

$$\exp(\langle\mu_i,\mu_j\rangle)\alpha\exp(\langle\mu_i,\mu_j\rangle - \frac{1}{2}\|\mu_i\|^2 - \frac{1}{2}\|\mu_j\|^2)$$

where $\alpha$ denotes proportionality, and this proportionality holds when $\|\mu_i\| = \|\mu_j\|$ and constant. We recognize the right hand side as the Gaussian kernel ([Keerthi and Lin](), 2003), which is always applicable. It simply remains to ask if the correction factor $\exp(-\frac{1}{2}\|\mu_i\|^2 - \frac{1}{2}\|\mu_j\|^2)$ between the two kernels is strongly concentrated. By hypothesis, we are near convergence, i.e. $\mu_i$ is approximately distributed in a Gaussian fashion. We know that high dimensional Gaussians are closely approximated by the uniform distribution on the hypersphere - that is, $\|\mu_i\|^2$ is strongly concentrated around a constant (1) - and thus, since the function in question is Lipschitz over the domain, $\exp(-\frac{1}{2}\|\mu_i\|^2 - \frac{1}{2}\|\mu_j\|^2)$ also concentrates in measure.

**Non-inclusion of the positive term** : If we instead have :

$$\mathbb{E}\log\sum_{X_j\in\mathcal{X},j\neq i}\exp(\langle\mu_i,\mu_j\rangle)$$

Then this is a **leave-one-out** evaluator of the kernel ([Barnard](), 2010). It is well known that in this case, the sum (upto some scaling, and with a different bandwidth than the original sum) again approximates the entropy, but with an error equal to (in expectation) the generalization error. Hence, these two analyses do not differ asymptotically.

**Non asymptotic convergence** : For this case, we bring back the temperature term and assume it to be set "correctly". Asymptotic results guarantee convergence for all temperatures, but in the nonasymptotic domain, this case is only analyzable in the low temperature limit, because under the non-low temperature scenario we have :

$$\mathbb{E}\log\sum_{X_j\in\mathcal{X}}\exp(\frac{\langle\mu_i,\mu_j\rangle}{\tau})$$

However, we recognize that the sum (after $\mathbb{E}\log$) is of a random variable which is the exponential of a Gaussian when $\mu_i,\mu_j$ are isotropic Gaussians. This is because $\langle\mu_i,\mu_j\rangle$ is the scaled projection of all $\mu_j$ on a fixed random vector $\mu_i$. This is a 1-dimensional gaussian, making the sum a sum of log normal random variables. While this can be approximately expressed via methods such as the Fenton-Wilkinson moment matching method, this is far less clean than the case of low $\tau$.

Instead, consider $\tau << 1$. We have that :

$$\mathbb{E}\log\sum_{X_j\in\mathcal{X}}\exp(\frac{\langle\mu_i,\mu_j\rangle}{\tau})$$

is approximately equal to :

$$\arg\max_j\frac{\langle\mu_i,\mu_j\rangle}{\tau}$$

We recall the definition of the Kozachenko Leonenko estimator : for a sample consisting of $X_1, X_2, \ldots, X_{N+1} \in \mathbb{R}^d$ drawn from an unknown distribution $P$, assuming $N > 1$, With $R_i = \min_{j\neq i}\|X_i - X_j\|_2, Y_i = N(R_i)^d, B_d$ the volume of the unit ball in $\mathbb{R}^d$, $\gamma$ the Euler-mascheroni constant $\approx 0.577$, an estimate of the entropy of the distribution is:

$$H(P) \approx \frac{1}{N+1}\sum_{i=1}^{N+1}\log Y_i + \log B_d + \gamma$$

We do not need to calculate $B_d$ and $\gamma$ as they are constants per instance. Rather, we observe that :

$$\log Y_i = d(\log N + \log R_i)$$

Changing notation from $X_i$ to $\mu_i$ for the purpose of unifying our derivations, note that $R_i$ is attained at the lowest value of $\|\mu_i - \mu_j\|_2$ i.e. at the highest value of $\langle\mu_i,\mu_j\rangle$ if $\|\mu_i\|, \|\mu_j\|$ are constants.

However, since we do not care about scaling or constant shifts (recall that optimizing a function $f$ is equivalent to working with $\lambda f + c$ for any $\lambda > 0$ and any constant c), we can express $\log R_i$ as :

$$\frac{1}{2} \times 2\log R_i = \frac{1}{2}\log R_i^2$$

$$= \arg\min_j\frac{1}{2}\log\|\mu_i - \mu_j\|^2$$

Expanding the norm squared, we get

$$\arg\min_j \frac{1}{2} \log(||\mu_i||^2 + ||\mu_j||^2 - 2\langle \mu_i, \mu_j \rangle)$$

We can use the fact $||\mu_i||^2, ||\mu_j||^2$ concentrate to approximate the above as

$$\approx \arg\min_j \frac{1}{2} \log(2 - 2\langle \mu_i, \mu_j \rangle)$$

Take the 2 in common, and note that this comes out of the log as $\log(ab) = \log a + \log b$. But we do not care about such constants as optimizing $f$ is equal to working with a linear transform of $f$ upto the learning rate, leaving us with

$$\arg\min_j \frac{1}{2} \log(1 - \langle \mu_i, \mu_j \rangle)$$

Finally, note that $\mu_i, \mu_j$ are high dimensional isotropic Gaussians of dimension $d$. Thus, $\langle \mu_i, \mu_j \rangle$ is a zero mean univariate Gaussian of variance $\frac{1}{d}$, i.e. with high probability, we have that :

$$\langle \mu_i, \mu_j \rangle << 1$$

Allowing us to replace the above term with (applying the identity that for $x << 1, \log(1+x) \approx x$) :

$$\arg\min_j \frac{1}{2}(-\langle \mu_i, \mu_j \rangle)$$

Swapping argmin with argmax and positive with minus we finally get :

$$\arg\max_j \frac{1}{2}\langle \mu_i, \mu_j \rangle$$

Which is (upto scaling by $\frac{2}{\tau}$) the desired non-asymptotic approximation.

## C   Extra Results - Penn Tree bank

Here, we add results on the penn tree bank dataset in table 11. Only a subset of all models that do not use fixes specific to the LSTM architecture were considered, since we had to adapt the tricks to transformers. Our base model is Transformer-XL.

## D   GPT-4 validation

We asked GPT-4 to validate the choices generated by our models, using the following prompt : "You are a human asked to choose between more realistic sentences. Among the following sentences, which

is the most consistent and high quality semantically, grammatically, and linguistically ?". This yielded the following results. Note that GPT-4 actually yielded more strong results towards our model than the human evaluators.

| Candidate | CEAE | BN-VAE | VAE |
|-----------|------|--------|-----|
| Yahoo | **55.7** % | 28.9 % | 15.4 % |
| Yelp | **67.3** % | 24.5% | 8.2 % |

Table 7: Frequency of choice among generated samples among **LSTM** models by GPT-4.

| Candidate | CEAE | BN-VAE | VAE |
|-----------|------|--------|-----|
| Yahoo | **59.4** % | 25.1 % | 15.5 % |
| Yelp | **52.0** % | 30.4 % | 17.6 % |

Table 8: Frequency of choice among generated samples among **transformer** models by GPT-4.

## E   BERT validation

We randomly selected 1000 sentences from Yelp and Yahoo corpora. This yields a total of $1000 \times 999 \times \frac{1}{2}$ pairs of sentences, each of which yields an inner product similarity. The corresponding similarities were computed from BERT (Devlin et al., 2018). We can then compare the quality of embeddings from BN-VAE, VAE, CEAE with respect to BERT over three choices of correlation metrics : Pearson (linear), Spearman (rank-based) and Kendall (pairwise). The results are as follows.

| Metric | CEAE | BN-VAE | VAE |
|--------|------|--------|-----|
| Pearson | **0.46** | 0.35 | 0.33 |
| Spearman | **0.76** | 0.70 | 0.68 |
| Kendall | **0.82** | 0.75 | 0.70 |

Table 9: Yahoo dataset, comparison of correlation metrics w.r.t BERT.

## F   The Layer norm case

Under layer normalization (Ba et al., 2016), the normalization is not done over a minibatch but over a layer. A consequence of this is the fact that the resulting latents form a hyperspherical space, i.e. $||z|| = c$ where $c$ is a constant. Now, we know that :

$$\arg\max H(z), ||z|| = c$$

is the uniform distribution over the hyperspherical shell of radius $c$. It is also well known that in high dimensions, the multivariate isotropic Gaussian has almost all of its support (Vershynin, 2018)

| Metric | CEAE | BN-VAE | VAE |
|---|---|---|---|
| Pearson | **0.58** | 0.45 | 0.48 |
| Spearman | **0.82** | 0.78 | 0.75 |
| Kendall | **0.86** | 0.75 | 0.79 |

Table 10: Yelp dataset, comparison of correlation metrics w.r.t BERT.

concentrated around the hyperspherical shell of $\sqrt{d}$, where $d$ is the dimensionality. Hence, under the entropic regularizer, Layernorm approximates a hyperspherical distribution which in turn approximates a Gaussian.

Although this direction is promising, we were only able to get our method (CEAE) operational under these circumstances, while BN-VAE did not converge. In the absence of a full evaluation, the Layernorm case is as of now inconclusive.

# G   Statistical Analysis

Here we present results (in tables 12 and 13) on the standard deviation and confidence intervals on Yahoo and Yelp for selected models that perform well in terms of point estimates and rank at or near the top. These are the $\delta$-VAE, BN-VAE and our method (CEAE). Bolding indicates passing a $t$-test with $p < 0.01$.

| Model | Rec | GPT2-F | L-F | L-R |
|---|---|---|---|---|
| VAE | 42.8 | 62.7 | 138.5 | 211.7 |
| $\beta$-VAE (0.4) | 43.2 | 64.2 | 140.5 | 225.8 |
| cyclic | 41.2 | 70.2 | 141.8 | 230.2 |
| FB (7) | 41.7 | 56.5 | 130.2 | 215.6 |
| $\delta$-VAE (0.1) | 42.0 | 58.9 | 132.9 | 208.3 |
| BN-VAE (0.7) | 41.5 | 54.6 | 135.6 | 202.8 |
| CEAE | 39.2 | 54.0 | 127.7 | 195.4 |

Table 11: Language modeling results on Penn Tree Bank using a 6-6 encoder-decoder transformer model with hyperparameters based on the Transformer-XL setup for Penn Tree Bank as released by the authors.

| Model | Rec | GPT2-F | L-F | L-R |
|---|---|---|---|---|
| $\delta$-VAE (0.1) | 327.2-328.0(0.3) | 120.8-124.0 (1.1) | 129.5-131.8 (0.8) | 169.5-174.2 (1.5) |
| BN-VAE (0.7) | 317.7-319.5(0.6) | 108.6-110.0 (0.5) | 122.9-126.7 (1.2) | 166.2-170.1 (1.3) |
| CEAE | **316.0-317.5** (0.4) | **107.2-108.4** (0.4) | **118.5-122.3** (0.3) | **162.2-164.7** (0.8) |

Table 12: 5 to 95 percentile confidence intervals and standard deviations in brackets on Yahoo.

| Model | Rec | GPT2-F | L-F | L-R |
|---|---|---|---|---|
| $\delta$-VAE (0.1) | 354.3-358.7(1.4) | 93.2-97.0(1.2) | 83.5-85.0(0.5) | 106.7-113.5 |
| BN-VAE (0.7) | 345.2-347.6 (0.8) | 89.0-91.2 (0.7) | 79.4-81.2 (0.6) | **107.2\*-112.4\*** (1.7) |
| CEAE | **340.9-343.5** (1.2) | **87.2-88.6** (0.5) | **76.8-79.7\*** (1.0) | 108.4-112.9 (1.6) |

Table 13: 5 to 95 percentile confidence intervals and standard deviations in brackets on Yelp. * indicates statistically insignificant best result.