# OpenReview forum: "Contrastive Deterministic Autoencoders For Language Modeling"
_EMNLP/2023/Conference — EMNLP 2023 Findings_

### Official Review · Reviewer_1NbG · 2023-07-31

**Soundness:** 2

**Excitement:**

3: Ambivalent: It has merits (e.g., it reports state-of-the-art results, the idea is nice), but there are key weaknesses (e.g., it describes incremental work), and it can significantly benefit from another round of revision. However, I won't object to accepting it if my co-reviewers champion it.

**Missing References:**

Bowman et al. (2016): https://aclanthology.org/K16-1002.pdf

**Paper Topic And Main Contributions:**

This paper proposes a deterministic autoencoder for language modelling as an alternative to variational autoencoders (VAE) of text data, which are known to exhibit the so-called posterior collapse problem. Their methodology builds on the work of Ghose et al. (2020), but regularizes the model's objective function with a contrastive learning term. The latter is shown to approximate an entropic regularizer for Gaussian distributions in the limit of high dimensions.

The authors trained their model on three public datasets (PTB, Yelp and Yahoo) and showed that their models outperform different VAE models on both text generation and downstream tasks. The authors additionally showed their results to be consistent across different sequence-processing architectures (i.e. LSTMs vs Transformers), and even considered pretrained language models as the encoder-decoder pair. See however our questions below on the comparison with the work of Li et al. (2020).

*Links to References:*

Ghose et al. 2020: http://proceedings.mlr.press/v124/ghose20a/ghose20a.pdf

Li et al. 2020: http://aclanthology.lst.uni-saarland.de/2020.emnlp-main.378/

**Questions For The Authors:**

(1)  In line 933-934 is written that the chosen $\lambda$ parameter [of the Optimus model by Li et al. (2020)] corresponds to models with the lowest perplexity. Why did you choose this criteria? Why not to choose $\lambda$ values for which the MI in Optimus is large? Note that the reconstruction error for larger $\lambda$ values reported by Li et al. (2020) are sometimes better than those obtained with CEAE.

(2) When training CEAE for comparison against Optimus, was CEAE pretrained on English Wikipedia as Optimus? The experimental section about the comparison with Optimus should probably be extended. And CEAE models should probably be compared with Optimus models with other $\lambda$ values

(3) Can you please explain in more detail whether the scale of the variance (i.e. $c$) in CEAE is annealed to zero?

*Link to References*

Li et al. (2020): http://aclanthology.lst.uni-saarland.de/2020.emnlp-main.378/

**Reasons To Accept:**

The paper represents a first successful attempt to employ deterministic autoencoders for language modelling.

**Reasons To Reject:**

(1) The paper needs rewriting. It doesn't seem to be written for an NLP audience:

- When reading the paper one gets the impression that the authors have not considered the large literature on the posterior collapse problem of VAE **specific to language modelling**.  For example, posterior collapse in VAE models for language modelling (LM) was identified already in the work of Bowman et al. (2016), which to my knowledge was the very first attempt to use VAE with text data. This work is however not cited by the authors, but several more recent works on VAE for images are cited instead. Unfortunately, the lack of a related work section exacerbates this feeling. For example, the paper relies on contrastive regularization but does not cite prior work on contrastive learning and NLP (see e.g. the review in Zhang et al. 2022).

- Some of the notation used by the authors seems to have been taken directly from a VAE paper on images. Look for example at Eq. 4. Is the reconstruction loss used by the authors truly the mean squared error between word embeddings? or is it instead the log likelihood functions of a categorical distribution over some vocabulary (i.e. cross-entropy)? If it is indeed a mean squared error, then I don't understand the scales of the reconstruction values in Tables 1 and 2.

- It's not so clear the relevance of this work when compared with the current state of affairs in NLP. I believe a paper on sentence representation nowadays should try to connect with the current paradigm of foundational models and their most pressing problems. For example, how would large pretrained language models benefit from inferring continuous representations of sentences? The authors make little to no reference to foundational models. Their only motivation seems to be to bypass the posterior collapse problem.

- Somewhat more general, the paper relies on information theoretic concepts to infer its representations. Yet no connection is made with all previous literature on VAE, the ELBO and the mutual information between data and representation (see e.g. Zhao et al. 2018 in general or Zhao T. et al. 2018 and Fang et al. 2019 which are specific to LM). It'd be nice to relate their approach to previous work, since both ultimately rely on maximizing the mutual information between data and representation

- Finally, the presentation of VAE focuses only on the cases where the prior is an isotropic Gaussian. But the prior needs not be an isotropic Gaussian. Indeed researchers in NLP have considered training the prior distribution (Fang et al. 2019) or learning structured, discrete representations (e.g. Zhao T. et al. 2018).

(2) See also Question 1 below.

*Links to references*

Bowman et al. (2016): https://aclanthology.org/K16-1002.pdf

Zhang et al. (2022): https://aclanthology.org/2022.naacl-tutorials.6.pdf

Li et al. (2020): http://aclanthology.lst.uni-saarland.de/2020.emnlp-main.378/

Zhao et al. (2018): https://arxiv.org/pdf/1706.02262.pdf

Zhao T. et al. (2018): https://aclanthology.org/P18-1101.pdf

Fang et al. (2019): https://aclanthology.org/D19-1407.pdf

**Reproducibility:**

3: Could reproduce the results with some difficulty. The settings of parameters are underspecified or subjectively determined; the training/evaluation data are not widely available.

**Reviewer Confidence:**

3: Pretty sure, but there's a chance I missed something. Although I have a good feel for this area in general, I did not carefully check the paper's details, e.g., the math, experimental design, or novelty.

**Typos Grammar Style And Presentation Improvements:**

The paper is difficult to read because of the lack of consistency and formality in notation:

- it seems the authors use capital and lower case letters to denote both random variables and their values interchangeably. See e.g. Eq. 1 ($\mu$ as a realization) vs equation below line 251 ($\mu$ as a random variable) vs that below line 255 ($Z$ as a random variable). Then again in the Equations below lines 306, 327 and 350.

- it seems the authors use the index "i" to denote instances of the dataset and "j" to denote the components of the representations. But then it's not clear what the difference is, notation-wise, between the expectations in Eqs 8 and e.g. the expectation below line 251. Sometimes these indices are even used together, as in line 227.

[EDIT: I removed the comment regarding including comparison against Sanchez et al. 2023 (https://aclanthology.org/2023.acl-long.263.pdf), for the latter was officially published after the EMNLP deadline]

---

> ### Author Rebuttal · Authors · 2023-08-29
>
> Dear reviewer,
>
> We thank you for the review. Our reply follows.
>
> First, in the reasons to reject :
>
> 1.  Our paper is indeed somewhat written with a general ML audience in mind - hence the choice of the track “Machine learning for NLP”. We will certainly add NLP-specific works for VAEs in the related work, and you are correct that our primary inspiration was from the image-based works and trying to extend them further. We did not have enough space for a dedicated related works section, but we do have prior work section, which is of the same purpose. We agree that perhaps too much space went into the generic AE literature over the specific case of AEs for NLP, however, we did not wish to confuse those without background in autoencoders, which have recently become less popular.
>
> 2. This section generically sets up the AE. No, the reconstruction loss here is not directly used in the NLP loss (Reconstruction) in the experiments, for which indeed the likelihood is used. We apologize for this - this is meant as a general overview of AE, which generally utilizes the square loss for most circumstances outside NLP. We should have clarified that this carries over to likelihood in the sense that under Gaussian likelihood, the square loss can be interpreted as a log likelihood term.
>
> 3. Correct - the large advances in foundational models does pose problems to the relevance of VAE and older models in general. Our work tries to upgrade the VAE to the modern world by seeing if the techniques hold up with relatively larger encoder-decoder pairs like BERT. We agree that more work needs to be done to see if these techniques scale up to large, GPT-scale models, but we did not possess the compute for such.
>
> 4. Thank you for the references. Although, we note that we had previously seen Zhao 2018 and cited it (InfoVAE). We will integrate these.
>
> 5. Right, we only chose the Gaussian as training in this case for a larger model, in the form of OPTIMUS had been attempted before and thus had some foundation.
>
> Coming to the questions :
>
> A. Agreed, this is arbitrary, and was chosen as perplexity is a popular metric of model quality. If you prefer we can take the best $\lambda$ for the reconstruction instead.
>
> B. Yes. In the OPTIMUS part of the results, only the batch norm and the way the loss is crafted changes and nothing else is, for reasons of consistency.
>
> C. As stated in A.1, $c$ is set to zero after a threshold of epochs, and not approaching zero.
>
> We will take your presentation and grammar-based issues into account, specifically re : variables and instantiations, and inconsistent capitalizations. And yes, line 251 is somewhat confusing in its indexing. We will make $i,j$ consistent.

---

### Official Review · Reviewer_1cF2 · 2023-08-11

**Soundness:** 4

**Excitement:**

3: Ambivalent: It has merits (e.g., it reports state-of-the-art results, the idea is nice), but there are key weaknesses (e.g., it describes incremental work), and it can significantly benefit from another round of revision. However, I won't object to accepting it if my co-reviewers champion it.

**Paper Topic And Main Contributions:**

The paper delves into the challenge of posterior collapse in variational autoencoders (VAEs) when applied to language modeling.

**Main Contributions**:
1. **Introduction of the Contrastive Entropic Autoencoder (CEAE)**: A structure that leverages contrastive learning to adjust entropy, aiming to mitigate the posterior collapse problem inherent in VAE-based language models.
2. **Highlighting the limitations of existing VAE-based models**: The paper emphasizes that while many VAE models suffer from posterior collapse, the methods proposed to address this often negate the benefits of VAEs.
3. **Proposing a solution that retains VAE benefits**: The CEAE is designed to counteract posterior collapse without sacrificing the advantages offered by VAEs.
4. **Utilization of contrastive learning**: The CEAE employs contrastive learning specifically for Batch Normalization-based VAEs.
5. **Comparative analysis**: The paper showcases the superiority of the proposed method by comparing it with various VAE-based language models and state-of-the-art models like GPT-2 and BERT.
6. **Examination of the latent space**: The authors explore the latent space through interpolation, providing deeper insights into the model's capabilities.
7. **Emphasis on the importance of addressing posterior collapse**: The paper underscores the criticality of mitigating posterior collapse in VAE architectures, especially in the context of NLP.

**Questions For The Authors:**

- **Question A**: The paper mentions that BERT+GPT2 was selected due to the lack of an encoder-decoder structure that mirrors an autoencoder. However, models like BART and T5, which are based on encoder-decoders, have demonstrated state-of-the-art performance across various tasks. Why weren't experiments conducted using these models?

- **Question B**: The term "outsized benefits in NLP" mentioned in line 66 is somewhat vague. Could the author provide a more detailed explanation or context regarding this phrase?

- **Question C**: In line 270, the paper states, "Under this condition of fixed mean and variance, it is known that the entropy H is maximized iff z is distributed as a Gaussian." Could the author elaborate on the underlying rationale for this statement?

- **Question D**: Typically, layer normalization is favored over batch normalization for text data. While this is acknowledged in Appendix F, the paper also mentions that the conclusion was not definitive without a "full evaluation." Could the author clarify what constitutes a "full evaluation" in this context?

- **Question E**: In Table 6, phrases like 'with the beans' and 'with the salt' appear during interpolation, even though they don't seem to have a direct correlation with the source sentence. Could this phenomenon be indicative of posterior collapse?

**Reasons To Accept:**

- **Comprehensive Review of Existing Studies**: The paper provides a well-organized overview of existing studies addressing the posterior collapse issue.
- **Innovative Approach**: The build-up to address posterior collapse is distinct from existing methods, offering a fresh perspective on the problem.
- **Effective Model Structure**: The author introduces a model structure that adeptly integrates Contrastive Learning and Batch Normalization, showcasing its potential in addressing the challenges of VAE-based language models.
- **Reproducibility**: Essential details for reproducing the experiments and results are meticulously organized in the Appendix, promoting transparency and facilitating further research.
- **Latent Space Exploration**: The paper's examination of the latent space through interpolation provides deeper insights into the model's capabilities and behavior.

**Reasons To Reject:**

- **Not Tailored for NLP Audience**: The paper, while technically sound in some areas, does not seem to be specifically tailored for an NLP audience, which might limit its impact and relevance in the community.
- **Insufficient Experiments on Posterior Collapse**: While the author emphasizes the mitigation of posterior collapse in existing VAE-based language models, there seems to be a lack of comprehensive experiments and analysis to support this claim.
- **Unclear Advantages Over Existing Models**: The paper does not clearly delineate the advantages of the proposed model compared to existing solutions for posterior collapse, leaving readers uncertain about its unique contributions.
- **Questionable Large Language Model Comparison**: The experiments conducted for comparing large language models appear to be inadequate. Labeling BERT as a "Large Language Model" seems outdated given the recent advancements in the field.
- **Over-reliance on BN-VAE**: A significant portion of the paper is based on BN-VAE (Batch Normalization VAE). However, the rationale for its specific use and its significance in the context of the paper's objectives is not adequately explained.

**Reproducibility:**

5: Could easily reproduce the results.

**Reviewer Confidence:**

3: Pretty sure, but there's a chance I missed something. Although I have a good feel for this area in general, I did not carefully check the paper's details, e.g., the math, experimental design, or novelty.

**Typos Grammar Style And Presentation Improvements:**

- **Section Organization**: The experimental design for human evaluation, mentioned around line 480, would be more appropriately placed in the "4. Experimental Details and Methodology" section rather than in "5. Results". This reorganization could enhance the flow of the paper and improve readability for the audience.

---

> ### Author Rebuttal · Authors · 2023-08-29
>
> Dear reviewer,
>
> Thanks for your review ! First, we will address the reasons to reject, in sequence.
>
> 1. We believe our paper, though somewhat more ML-oriented than purely NLP, has ample relevance towards NLP, and we have submitted specifically to the Machine learning for NLP track for this reason.
>
> 2. We would appreciate knowing a specific experiment that you feel is lacking - this comment is a bit vague to work with. We do perform posterior collapse experiments as every test on latent representation quality, interpolation, etc. measures the degree of collapse.
>
> 3. Again, the advantages for posterior collapse generally arise in representation quality and the reconstruction error - both of which do provide advantages for our model.
>
> 4. We have used GPT-2 in conjunction with BERT, which is a reasonably large model (we are aware that there have been massive advancements in this area recently, and are trying to update accordingly)
>
> 5. BN-VAE is relevant due to its utilization of batch norm with VAEs for NLP, and its (at the time) strong results with smaller architectures.
>
> Then, regarding the questions :
>
> A. Please see our response to reviewer kNe8. The fine-tuning of OPTIMUS does not carry over to T5 cleanly in a way that allows convergence.
>
> B. Certainly. Text relies more heavily on autoregressive models than images. As such, since posterior collapse afflicts autoregressive models more, fixing it is more likely to impact text performance than image performance.
>
> C. It is a standard result in information theory. The result is one of the Gibbs distributions, i.e. maximum entropy distributions. Basically, with some moments fixed, the max entropy distribution is known. With fixed mean and variance, it is Gaussian. Under absolute deviation constraint, it is the Laplacian, and so on.
>
> D. We could not get the BN-VAE models to converge under the LayerNorm scenario, so we could not know if they truly perform worse, or if we did not give them a fair chance.
>
> E. We do not see why that would follow. This dataset is about reviews of restaurants in yelp, and it is reasonable to have food-related items appear in the interpolation without necessarily implying that posterior collapse has occurred.
>
> We will also take your suggestion for changing the placement of the human eval section into account.

---

### Official Review · Reviewer_kNe8 · 2023-08-13

**Soundness:** 2

**Excitement:**

3: Ambivalent: It has merits (e.g., it reports state-of-the-art results, the idea is nice), but there are key weaknesses (e.g., it describes incremental work), and it can significantly benefit from another round of revision. However, I won't object to accepting it if my co-reviewers champion it.

**Paper Topic And Main Contributions:**

This paper introduces a framework that adapts deterministic autoencoders for language modeling. This method employs contrastive learning, instead of the conventional KL loss, to make the posterior distribution Gaussian and hence avoid the posterior collapse.

**Reasons To Accept:**

The method consistently outperforms other VAE-based methods in two different datasets and works with different architectures (LSTM & Transformer). The Gaussian latent space, resulting from the autoencoder training, allows linear interpolation between sentences, which could be useful for future text generation or interpretation research.

**Reasons To Reject:**

It's unclear how the proposed method is superior to the conventional language modeling objective. Especially, a T5-style model training with its standard objective should be included as a baseline since it also uses an encoder-decoder architecture. Last, as language modeling is shifting to large scale pretraining, the method's scaling capability remains a question.

**Reproducibility:**

4: Could mostly reproduce the results, but there may be some variation because of sample variance or minor variations in their interpretation of the protocol or method.

**Reviewer Confidence:**

2: Willing to defend my evaluation, but it is fairly likely that I missed some details, didn't understand some central points, or can't be sure about the novelty of the work.

---

> ### Author Rebuttal · Authors · 2023-08-29
>
> Dear reviewer,
>
> Thank you for the review. T5 (along with some other models, e.g. BART) does have encoder-decoder structure but no latent space structure like a VAE, which means it is not directly comparable (this is noted in the initial parts of the OPTIMUS paper). It can be made comparable via introduction of a suitable latent space, just as we fuse BERT, GPT-2 and a latent space to create the OPTIMUS-VAE. We were unable to replicate the steps done for OPTIMUS on the T5 model to convert it to a VAE with satisfactory convergence, however, and we suspect a different approach is required. (The metrics of T5 without a latent space cannot be directly compared.)
>
> That said, we view BERT / GPT-2 as language models of comparable “power” to T5 in terms of encoding and decoding respectively, and we believe that it should suffice. Further, note that decoder-only models (unlike encoder-decoder models) are increasingly more prominent, and as such, pairing encoders and decoders is likely to be the way moving forward to convert large models to large VAEs, instead of inserting a latent space in an already existing encoder-decoder pair.
>
>
> We are keenly aware that the advent of large models will change the NLP landscape along with pre-training. But the comparison with BERT-GPT-2 models is meant to alleviate this concern and show VAE models can scale up reasonably well in the presence of large and pre-trained models. Indeed, note that the scaling procedure simply joins any two encoders and decoders of arbitrary size.
>
> As to the superiorities of a structured latent space over conventional LM, there are multiple positives. For instance, interpolation with a known latent space can be done better exploiting the geometry (i.e., geodesic interpolation) over linear interpolation. This interpolation can then give rise to superior generations from the interpolated representation. This empirically manifests not just in better generation, but better downstream classification accuracy, as we show in our experiments. Text generation is also easier as there is a known, closed-form source distribution to sample from.

---

### Meta-Review · Area_Chair_Q2km · 2023-09-19

**Recommendation:** 2

**Metareview:**

This paper proposes a deterministic autoencoder for NLP, as an alternative to VAE. The key idea is to introduce a contrastive learning loss to approximate entropy regularization. The paper also presents experiments on a variety of model architectures and downstream tasks.

Overall, reviewers have divided view on soundness while show excitement in this paper's approach.
Reviewers like the novel technical method, and the strong empirical performance on various architectures. However, reviewers also point out missing discussion with vast past literature on VAE for text, use of contrastive loss in NLP, analysis of mode/posterior collapse issues in VAE, isotropic Gaussian prior versus other priors (e.g. mixture-of-Gaussian). Reviewers also have concerns about the paper's reference to "large language models" since BERT, GPT-2 are no longer considered large nowadays.

---

### Decision · Program_Chairs · 2023-10-07

**Decision:**

Accept-Findings

**Comment:**

This paper proposes a deterministic autoencoder for NLP, as an alternative to VAE. The key idea is to introduce a contrastive learning loss to approximate entropy regularization. The paper also presents experiments on a variety of model architectures and downstream tasks.

Overall, reviewers have divided view on soundness while show excitement in this paper's approach.
Reviewers like the novel technical method, and the strong empirical performance on various architectures. However, reviewers also point out missing discussion with vast past literature on VAE for text, use of contrastive loss in NLP, analysis of mode/posterior collapse issues in VAE, isotropic Gaussian prior versus other priors (e.g. mixture-of-Gaussian). Reviewers also have concerns about the paper's reference to "large language models" since BERT, GPT-2 are no longer considered large nowadays.